# Edible Coatings Containing Oregano Essential Oil Nanoemulsion for Improving Postharvest Quality and Shelf Life of Tomatoes

**DOI:** 10.3390/foods9111605

**Published:** 2020-11-04

**Authors:** Annachiara Pirozzi, Vittoria Del Grosso, Giovanna Ferrari, Francesco Donsì

**Affiliations:** 1Department of Industrial Engineering, University of Salerno, via Giovanni Paolo II, 132, 84084 Fisciano, SA, Italy; apirozzi@unisa.it (A.P.); gferrari@unisa.it (G.F.); 2ProdAl Scarl, via Giovanni Paolo II, 132, 84084 Fisciano, SA, Italy; v.delgrosso@prodalricerche.it

**Keywords:** sodium alginate, calcium chloride, oregano essential oil, antimicrobial agent, response surface methodology, preservation, storability

## Abstract

Edible coatings have attracted significant interest in maintaining quality and improving the shelf life of fresh fruit and vegetables. This study aimed to improve tomato storability by using edible coatings, based on alginate cross-linked with calcium chloride, and containing an oregano essential oil (OEO) nanoemulsion as a natural antimicrobial. The coating formulations were preliminary optimized in terms of alginate and calcium chloride concentrations, using response surface methodology, to obtain a thin (~5 µm) and uniform layer on the tomatoes surface. The optimized coating (prepared using sequential dipping in a 0.5% *w*/*w* sodium alginate solution and in a 2.0% *w*/*w* calcium chloride solution) was enriched by incorporating an OEO nanoemulsion, formulated with lecithin as a natural emulsifier, at an OEO concentration of 0.17% *w*/*w* in the alginate solution. The nanoemulsion did not significantly affect the coating thickness and uniformity but improved the wettability of the tomato skin. More specifically, the alginate-based edible coatings exhibited a strong interaction with the hydrophobic tomato skin surface (higher than water), promoting surface adhesion. The addition of OEO nanoemulsion in the coating, by providing more hydrophobic sites, further improved the wetting capability and adhesion of the coating solution on the tomato surface. The developed edible coatings successfully contributed to prolonging the tomato shelf life, by reducing the growth of the endogenous microbial flora (total microbial load, yeasts, and molds) over 14 days at room temperature in comparison with the control, with significantly better performances for the edible coating containing the OEO nanoemulsion.

## 1. Introduction

Edible coatings represent a method traditionally used to extend the shelf life and maintain the physical and gustatory characteristics of fresh produce [1]. The recent introduction of food-grade materials with more advanced properties, as well as of novel deposition methods, has enabled the application of edible coatings in extremely thin layers on the food surface, providing an invisible physical barrier to oxygen, external microbial contamination, and moisture absorption/desorption in food, without affecting its organoleptic properties [2,3,4]. More recently, the use of edible coatings has been further promoted by the coating functionalization with bioactive compounds, such as natural antimicrobial compounds, antioxidants, minerals, and vitamins, which contribute not only to improving safety and preserving the quality of food but also to delivering health benefits to the consumer [5,6]. Because of their high efficiency in reducing the occurrence of deteriorative processes, antimicrobial coatings have been widely studied for extending the shelf life of fish and meat products, as well as for high value-added fruit and vegetables [3].

Edible coatings can be applied to the food surfaces by various methods, including spraying, dipping, spreading, and thin-film hydration [7]. Numerous studies have been conducted to date to assess the effect of edible coatings on preserving the quality and increasing the shelf life of fresh/fresh-cut fruit [4]. The coating efficiency in protecting food products was reported to depend on (i) the method of application; (ii) the nature of the coating ingredients and their concentrations [8], (iii) the uniformity of wetting and spreading on the surface and on the adhesion, cohesion, and durability [9]; and (iv) the capability to act as barriers against water or oils permeation, and gas or vapor transmission [9,10].

Alginates are widely used in edible coatings because of their wide availability and regulatory status. The U.S. Food and Drug Administration (FDA) classifies food-grade sodium alginate as generally regarded as safe (GRAS) [11] and lists its usage as an emulsifier, stabilizer, thickener, and gelling agent [12]. The European Commission (EC) lists alginic acid and its salts (E400–E404) as authorized food additives [13]. Alginates are anionic polymers composed of mannuronic and guluronic acid, which are recovered from various species of brown seaweeds via alkaline extraction, followed by precipitation with either sodium or calcium chloride [14]. They have been widely used, in combination with different ingredients and bioactive compounds, to fabricate active coatings. Moreover, alginates can be cross-linked by divalent ions, such as Ca^2+^, to develop a gel, which can be formed also as a thin film or coating [15]. For example, the addition of an immersion step into a calcium solution led to a smoother and more homogeneous alginate gel layer on both very hydrophilic and hydrophobic fruit surfaces [16]. Calcium chloride alone has also been found to be important in fruit and vegetable preservation: fruit and vegetables dipped in calcium chloride solutions decreased postharvest softening and decay when combined with heat treatment [17].

Recently, the incorporation of essential oils (EOs) as natural antimicrobial agents in edible coatings has received increasing attention to control the decay and extend the storage life of perishable foods [18,19,20]. Oregano essential oil (OEO), which is rich in the GRAS monoterpene carvacrol, has excellent antimicrobial properties and inhibits the growth of both Gram-positive and Gram-negative bacteria and fungi [21].

This work addressed the development of an active coating for tomatoes, which, according to FAO, are the second most abundantly consumed fresh produce worldwide [22]. The shelf-life extension of tomatoes is of great industrial relevance, especially to compensate for poor logistics from rural and remote areas of cultivation, hence contributing to food waste reduction. In this work, coating formulation, based on sodium alginate cross-linked with calcium chloride at different concentrations and dipping time, was optimized as a function of coating thickness when applied on tomato fruit. Subsequently, the optimized coatings were functionalized with the addition of OEO nanoemulsions for the inhibition of the growth of the endogenous flora, while preserving the quality properties of the tomatoes during refrigerated shelf life.

## 2. Materials and Methods

### 2.1. Materials

Fresh cherry tomatoes (*Solanum lycopersicum cerasiforme*) were purchased from a local supermarket (Salerno, Italy), and stored at 4 °C until used. Before each experiment, tomatoes were classified according to the maturity indices of tomato at different harvest stages, suggested by the U.S. Department of Agriculture [23], and those at a maturity stage number 5, characterized by a light red color (an aggregated fraction of the surface between 60% and 90% shows pinkish-red or red color), similar shape and without any physical damage or visible signs of disease were subjected to the experimental treatments.

Sodium alginate (Sigma-Aldrich Chemie GmbH, Steinheim, Germany), calcium chloride (Thermo Fisher GmbH, Kandel, Germany), oregano essential oil (kindly supplied by Frey & Lau GmbH, Henstedt, Ulzburg, Germany) and lecithin (soy lecithin Solec IP, Milan, Italy) were used in coating formulations. Buffered peptone water, plate count agar, and agar dichloran rose bengal (all acquired from VWR International, Milan, Italy) were used in microbiological viability tests.

All chemicals and solvents used in this study were purchased from Sigma Aldrich (Milan, Italy) unless otherwise specified.

### 2.2. Oregano Essential Oil Nanoemulsion Preparation

Nanoemulsions were produced by dispersing OEO in water, using lecithin as the emulsifier. A primary emulsion was prepared by mixing 0.5% *w*/*w* OEO with 0.5% *w*/*w* lecithin and then adding the mixture to a sterilized buffer (99% *w*/*w*) at pH 6.9 using a high-shear mixing (HSM) disperser (Ultra Turrax T25, IKA Labortechnik, Staufen, Germany) at 18,000 rpm for 5  min, based on previously optimized conditions [24]. The nanoemulsion was obtained by treating the primary emulsion by high-pressure homogenization (HPH) at 200 MPa for 5 passes, using an in-house developed unit, equipped with a 100 μm diameter orifice valve (model WS1973, Maximator JET GmbH, Schweinfurt, Germany) and an air-driven Haskel pump (model DXHF-683, EGAR S.r.l., Milan, Italy). The HPH system was equipped with two tube-in-tube heat exchangers with chilled water, placed immediately downstream of the pump and the orifice valve, to ensure that the processing fluid was quickly cooled down to 5 °C. The nanoemulsions were stored at ambient temperature (24 ± 1 °C) for 24 h before characterization.

### 2.3. Nanoemulsion Characterization

#### 2.3.1. Determination of Hydrodynamic Diameter, Polydispersity Index, and Zeta Potential

The OEO nanoemulsion hydrodynamic diameter and polydispersity index (PDI) were measured by dynamic light scattering (DLS) (Zetasizer Nano ZS, ZEN3600, Malvern Instruments Ltd., Malvern, UK) working at 633 nm and 25 °C, and equipped with a backscatter detector (173°). Measurements were carried out on undiluted samples loaded in DTS0012 square disposable polystyrene cuvettes. Samples for ζ-potential were, instead, diluted 1:10 in a buffer solution (pH = 7.4), using DTS1060 disposable folded capillary cells. The ζ-potential measurement was based on phase analysis light scattering (PALS) using the same Zetasizer apparatus, by measuring the electrophoretic mobility with He-Ne laser emitting at 633 nm and 4.0 mW power sources at 25 °C. Each measurement was replicated three times on independently prepared samples, with the means and the standard deviations being calculated.

#### 2.3.2. In Vitro OEO Nanoemulsions Release

OEO release rate from the nanoemulsions was determined in vitro using Franz static diffusion cells (SES GmbH-Analysesysteme, Bechenheim, Germany), consisting of a donor and a receptor compartment, separated by a cellulose acetate membrane (dialysis tubing cellulose membrane, MWCO = 14 kDa, from Sigma-Aldrich, Germany), as previously described [25,26]. The receptor compartment, agitated with a magnetic stirrer at 480 rpm, was equipped with a sampling tube through which periodic withdrawals were carried out and was thermostated at 37 °C with an external jacket. The donor compartment was filled with the OEO nanoemulsions (1 mL), while the receptor compartment was filled with 5 mL of a phosphate buffer solution (pH = 7.4). Samples (0.2 mL) were withdrawn at different times (from 0 to 96 h) from the receptor compartment and replaced with 0.2 mL of fresh buffer solution to retain the sink conditions in the system and keeping a constant volume. The reported results are means of two replicated measurements. The OEO concentration in the receptor compartment was indirectly assessed through the quantification of the total phenolic content (TPC) through the Folin–Ciocalteu method [27]. Briefly, 1.0 mL of the sample was mixed with 5.0 mL of Folin–Ciocalteu reagent (10% *v/v* in distilled water). Afterward, sodium carbonate (7% *w*/*v*, 4 mL) was added to the mixture. After 90 min storage in the dark, the absorbance values were measured at 765 nm by a UV–vis spectrophotometer (V-650 UV-VIS Spectrophotometer, Jasco, Pfungstadt, Deutschland). The OEO concentration was determined using a calibration curve, obtained from the linear fitting (A_765 nm_ = 963.34 × C_OEO_, *R*^2^ = 0.9993) of the measured absorbance (A_765 nm_) as a function of OEO concentration (C_OEO_ ranging from 200 to 5 mg/L) in a buffer solution (Appendix A).

The cumulative release *Q* of OEO in the receptor compartment was determined using Equation (1) [28]:(1)$$Q_{n}=C_{n}\cdot V_{2}+\overset{n-1}{\underset{i=1}{\sum }}C_{i}\cdot V_{i}$$
where *Q_n_* is the cumulative release at the *n*-th withdrawal, *V*_2_ is the volume of the receptor compartment, *C_i_* is the concentration of OEO in the *i*-th withdrawal, and *V_i_* is the corresponding withdrawn volume.

OEO effective diffusivity through the membrane (*Ɗ*) was, instead, calculated, using Equation (2), by linear fitting of the linearization (*Y*) of the concentrations measured in the cell as a function of time *t* [29]:(2)$$Y=\frac{1}{B}\cdot \ln\frac{C_{10}-C_{20}}{C_{1}-C_{2}}=Ɗ\cdot t$$

In Equation (2), *B* is the cell constant (1.04 cm^−2^), whose value was preliminarily determined using a compound of known diffusivity, such as KCl in a 0.1 M solution [25]; *C*_20_ is the initial concentration of OEO in the receptor compartment; *C*_2_ is the concentration of OEO in the receptor compartment at time *t*; *C*_10_ is the initial concentration of OEO in the donor compartment; and *C*_1_ is the concentration of OEO in the donor compartment at time *t*, determined through the mass balance of Equation (3).
(3)$$C_{10}\cdot V_{1}+C_{20}\cdot V_{2}=C_{1}\cdot V_{1}+C_{2}\cdot V_{2}+\overset{n-1}{\underset{i=1}{\sum }}C_{i}\cdot V_{i}$$

### 2.4. Preparation of Coating Solutions

Sodium alginate (SA) and calcium chloride coating solutions were prepared by separately dissolving sodium alginate and calcium chloride in distilled water, followed by stirring with a magnetic stirrer, at 70 °C and 25 °C, respectively, until the solutions became clear [30,31,32]. The solutions were then sterilized in an autoclave at 121 °C for 10 min. The active edible coating was prepared by mixing a sodium alginate solution of appropriate concentration with OEO nanoemulsion under magnetic stirring at a mass ratio of 2:1, which was selected from preliminary experiments to ensure a final alginate solution of 0.5% *w*/*w* (optimized concentration) and an OEO concentration of 0.17% *w*/*w*. All coating solutions were prepared in a laminar-flow hood (Faster Bio48/Faster UltraSafe, Milan, Italy) under aseptic conditions, using autoclave-sterilized Milli-Q water.

### 2.5. Coating Application

Tomatoes were withdrawn from cold storage (4 °C) and left to equilibrate at room temperature (24 ± 1 °C) for 2 h. Before coating deposition, the tomatoes were thoroughly washed with running water and manually dried with absorbent paper. Coatings were applied onto the fruit surface using the dipping technique, based on the complete immersion of the fruit in the coating solutions for 2 min, and then draining off the excess coating. To this end, tomatoes were pricked in their pith with a sterile pin, used to hold the tomatoes submerged for the desired dipping time in the alginate and calcium chloride solutions. Tomatoes dipped in sterilized Milli-Q water, also pricked in their pith with a sterile pin, were used as control samples. The coated tomatoes were drained and dried in a laminar-flow hood (24 ± 1 °C) in aseptic conditions for 3 h before further analysis.

### 2.6. Response Surface Methodology for Coating Thickness Optimization

Response surface methodology (RSM) for the design of experiments (DOE) was used to determine the optimum coating and cross-linking solution concentrations and dipping times for the deposition of the alginate coating layer. The response variable was the thickness of the coating layer, evaluated using a digital micrometer (see Section 2.7.1). Central composite face-centered design was applied in this study, using Design-Expert 11.1.2.0 software (Statease Inc., Minneapolis, MN, USA). Sodium alginate concentration (% *w*/*w*), calcium chloride concentration (% *w*/*w*), and dipping time (s) were selected as the independent variables to be optimized. The lowest and highest values of the independent variables (0.25 and 5.0% *w*/*w* for sodium alginate concentration, 0.25 and 7.0% *w*/*w* for calcium chloride concentration, and 10 and 300 s for dipping time) were selected based on preliminary laboratory tests. The values of the coating thickness, used as the response variable (*Y*_1_), were related to the coded variables (*X_j,j_*, with *i*, *j* = 1, 2, 3) by a second-order polynomial, reported in Equation (4).
(4)$$Y_{1}=\beta_{0}+\underset{j}{\sum }\beta_{j}X_{j}+\sum^{}\underset{i<j}{\sum }\beta_{ij}X_{i}X_{j}+\varepsilon$$

The coefficients of the polynomial model correspond to a constant term (*β*_0_), linear effects (*β_j_*), and pure second-order interaction effects (*β_ij_*), with *ε* being the model error.

The significant terms of the model were determined through analysis of variance (ANOVA) for the response variable, carried out through the Design-Expert 11.1.2.0 software. The adequacy of the model was assessed by determining the F-test value for the regression (*F_reg_*) and lack of fit, the determination coefficients (*R*^2^, *Adj-R*^2^, *Pred-R*^2^), and probability values (*p*-value). Numerical and graphical optimization techniques of the Design-Expert 11.1.2.0 software were used for the optimization of the coating thickness response.

The central composite design (CCD, expert 11.1.2.0) method was also used to determine the number of experiments to be evaluated for the optimization of the three independent and the response variables. The coded independent variables used in the RSM design are listed in Table 1.

### 2.7. Edible Coatings Characterization

#### 2.7.1. Thickness Determination

Once the coatings completely dried off, the edible coating layers formed on the tomato surface were manually removed for the determination of their thickness. Eight measurements at distinct points of the coating removed from each tomato were performed in duplicate, using a digital micrometer (Micromaster Capa µ System, TESA Technology Italia s.r.l., Milan, Italy), and the mean value was determined.

#### 2.7.2. Light Microscopy Analysis

The microscopic structure of the coating layer deposited on the tomato surface was observed in situ using an optical inverted microscope Nikon Eclipse (TE 2000S, Nikon instruments Europe B.V., Amsterdam, The Netherlands), with 10× and 20× objectives, coupled to a DS Camera Control Unit (DS-5M-L1, Nikon Instruments Europe B.V.) for image acquisition and analysis. Slices, 2 mm thick, were cut out of the coated tomatoes using a surgical blade in stainless steel and immobilized on a glass slide with a cover glass. Observations of the microstructure of the coating were obtained using the photographing option in the DS Camera Control Unit.

#### 2.7.3. Scanning Electron Microscopy (SEM) Analysis

The morphological features of coatings were analyzed also using scanning electron microscopy (SEM). Slices, 2 mm thick cut out of coated tomatoes, were fixed by immersion in a 2% (*v*/*v*) glutaraldehyde phosphate buffer solution. The buffer was removed, and the slices were osmotically dehydrated with ethanol solutions of increasing concentration (25%, 50%, 75%, and 100% (*v*/*v*)). Afterward, ethanol was removed with supercritical CO_2_ in a Quorum K850 critical point dryer (Quorum Technologies Ltd., London, UK), and the samples were metalized using the Agar Auto Sputter Coater 103A (Agar Scientific Ltd., Stansted, UK), before being analyzed in a high-resolution ZEISS HD15 Scanning Electron Microscope (Zeiss, Oberkochen, Germany).

### 2.8. Contact Angle Measurement and Wetting Properties

The contact angles (*θ*) of the film-forming solution on the tomato skin were measured by the sessile drop method [33], using a contact-angle meter (KSV Instruments LTD CAM 200, Helsinki, Finland), equipped with an image analysis software. Briefly, a drop of about 2 μL of the film-forming solution was gently dispensed on the tomato surface (placed on the instrument stand to be aligned horizontally in the contact point with the water drop) using a 500 μL syringe (Hamilton, Switzerland) with a 0.71 mm diameter needle. Measurements were made each 5 s for a total of 120 s to evaluate the contact angle changes over time. The contact angle measurements were carried out in open air at room temperature (24 ± 1 °C), in situ, using water as a control and the optimized film forming formulation, containing 0.5% *w/w* of sodium alginate with and without the addition of 0.17% *w/w* of OEO nanoemulsion. Contact angles were measured in triplicate, and the average contact angle was calculated as the mean value over the 120 s measurement.

From the contact angle measurement, the coatings’ capability to wet the tomato surface (wettability) was also determined. The surface tension (*γ_L_*) of the wetting solutions and the contact angle *θ* between the solutions and the solid surface (the tomato peel), experimentally measured, might give indications about their mutual interaction in terms of reversible work of adhesion, *W_a_*, according to Equation (5):(5)$$W_{a}=\gamma_{L}\cdot \left(1+\cos\theta \right)$$

At equilibrium, the contact angle can be considered as an intensive property because it is independent of the amount of liquid (gas or solid) used in the measurement. The contact angle of a liquid drop on a solid surface is defined by the mechanical equilibrium of the drop under the action of three interfacial tensions, namely, the solid–vapor (*γ_SV_*), solid–liquid (*γ_SL_*), and liquid–vapor (*γ_LV_*) tensions [34].

Wettability was evaluated by determining the spreading coefficient (W_s_) and the work of adhesion (*W_a_*) and cohesion (*W_c_*). *W_a_* and *W_c_*, representing the work of adhesion and cohesion, respectively, are defined in Equations (6) and (7) [34].
(6)$$W_{a}=\gamma_{LV}+\gamma_{SV}-\gamma_{SL}$$
(7)$$W_{c}=2\cdot \gamma_{L}$$

The equilibrium spreading coefficient (*W_s_*) is defined in Equations (8) and (9) [35]:(8)$$W_{s}=W_{a}-W_{c}=\gamma_{SV}-\gamma_{LV}-\gamma_{SL}$$
(9)$$W_{s}=\gamma_{L}\cdot \left(\cos\theta -1\right)$$

### 2.9. Microbial Population during Storage

The changes in the endogenous microbial flora of both control and coated tomatoes stored at 24 ± 1 °C for 14 days were determined through the count of total bacteria and yeasts and molds at storage days 1, 3, 7, 10, and 14.

The total bacterial count was measured using plate count agar (PCA) medium, according to the procedures described by ISO 4833-1/2013, while yeasts and molds were determined using agar dichloran rose bengal (DRBC), according to ISO 21527-1/2008. For each microbial determination, two tomatoes per coating formulation were analyzed by recovering the microbial load using a filter stomacher bag containing buffered peptone water (1:5 mass ratio), which was aseptically homogenized in a Stomacher 400 Circulator (Seward, FermionX, provided by VWR International PBI s.r.l., Milan, Italy). Decimal dilutions were prepared with the same diluent, and appropriate dilutions were included in culture media.

For total bacterial count, inoculated Petri dishes were incubated at 30 °C for 72 h; for molds and yeasts count, the Petri dishes were incubated at 25 °C for 120 h. After incubation, colonies were enumerated and results were reported as log CFU/g (colony forming units per gram) of sample. All tests were performed in triplicate, and for each sample, the evaluation was carried out in duplicate for each dilution and analyzed according to ISO TS 19036:2006. The sensitivity of the analysis was ≥1 log CFU/g.

### 2.10. Statistical Analysis

Treatments and analyses were performed in triplicate unless differently specified and the results were reported as means ± sd. The significance of differences between means was evaluated using a one-way analysis of variance (ANOVA), performed with SPSS 20 (SPSS IBM., Chicago, IL, USA) statistical package, and Tukey’s test (*p* < 0.05).

## 3. Results and Discussion

### 3.1. Physicochemical Stability of Nanoemulsions Encapsulating OEO

The OEO nanoemulsion was characterized in terms of droplet size distribution (hydrodynamic diameter and PDI values) and ζ-potential, with the values reported in Table 2, together with the nanoemulsion formulation. The prepared system exhibited a hydrodynamic diameter of 176.8 nm and a PDI of 0.26, suggesting that the nanoemulsion can be considered monodispersed and homogeneous, and hence less susceptible to physical instability than more polydisperse systems [36]. The OEO nanoemulsion also exhibited a negative surface charge, with an average ζ-potential of −23.7 mV. In general, high absolute values of ζ-potential (>30 mV) are associated with efficient stabilization mechanisms of the oil droplets through electrostatic repulsion [37]. Therefore, the obtained nanoemulsion did not show any sign of instability (e.g., creaming or oiling off) and always maintained a homogeneous appearance over a 1-month storage period at 4 °C. The nanoemulsion is likely stabilized by a combined effect of electrostatic and steric effects. The observed negative ζ-potential may be attributed to free fatty acids and other polar constituents present in the oil phase, which may adsorb at the interface of the emulsion droplets.

### 3.2. In Vitro Permeation OEO Nanoemulsion Formulation

The OEO release from the nanoemulsion was investigated in vitro in Franz cells. The release profile of OEO into the receptor compartment continuously increased from 6.2% after 1 h to 74.5% after 96 h (Figure 1). From the OEO release curve, the OEO apparent diffusivity through the membrane was determined, using Equations (2) and (3), to be 9.17 × 10^−7^ cm^2^/s.

A modified Langmuir equation was used to fit the release curve (Equation (10)), with the corresponding fitting parameters reported in Table 3. The selected empirical model is capable of adequately describing the observed experimental release data.
(10)$$y=\frac{a\cdot x}{1+b\cdot x}$$

### 3.3. Coating Characterization: Microstructure of Films

The morphological and structural properties of the edible coatings were characterized by optical microscopy and SEM, using coating solutions prepared with sodium alginate solutions at concentrations of 2.5% or 5% (*w*/*w*) and calcium chloride solutions at a concentration of 7% (*w*/*w*). The concentration of the coating solutions were selected to ensure that well-formed and easily removable coatings were obtained. The dipping time was set at 120 s.

The optical microscopy images (Figure 2), taken at different locations of the coating layer, showed that the sequential dipping of tomatoes in sodium alginate and calcium chloride solutions enabled the formation on the tomato surface of a continuous and homogeneous layer, of quite uniform thickness (with a variation typically contained within ±10%, as shown by the measurements of Figure 2, taken at different locations). This layer appeared as compact and did not show any visible separation in different phases or pore formation, nor any difference in optical density in the coating thickness.

In order to gain a better understanding of the effect of sodium alginate concentration, SEM analysis was also conducted to visualize the microstructure of the coating on the tomato peel surface. Figure 3 shows the SEM micrographs of the coating layers prepared under the same conditions as Figure 2. Apparently, the increase in sodium alginate concentration in the coating solution did not cause any remarkable structural change in the formed layer. Conversely, a smoother and more compact structure was observed for the 2.5% *w*/*w* sodium alginate solution than for the 5% *w*/*w*. Moreover, it can also be observed that, coherently with the optical microscopy, a higher concentration of sodium alginate caused a corresponding increase in film thickness, likely due to the increased solution viscosity [38]. In turn, this corresponds to a reduction of adhesion strength and the spreading coefficient of the coating solution and improvement of the cohesion coefficient [39]. It must be highlighted that sample preparation during SEM analysis might have led to the formation of some artifacts; thus, caution is suggested in the interpretation of these images.

### 3.4. Coating Thickness Optimization

The RSM was used to explore the effect of dipping time (*X*_1_) and concentrations of the coating and cross-linking solutions (*X*_2_ and *X*_3_) on the resulting thickness of the coating layers (*Y*_1_) (see Appendix A for details about experimental data for coating thickness). The 3D response surface plots, shown in Figure 4, graphically represent the multilinear regression of the coating thickness data as a function of composition and dipping variables (Equation (4)). The ultimate aim of the 3D response surface plots is to enable the prediction of the optimum values of the independent variables (solutions concentration and dipping time) that minimize the response variable (coating thickness).

The coating layer thickness resulted in depending strongly on the concentration of sodium alginate and calcium chloride (pictures of the samples are reported in Appendix A), whereas the dependence from dipping time, within the investigated range (10–300 s), was weak. Figure 5 explicitly reports the dependence of coating thickness from dipping time, for the highest concentrations of coating solutions tested (5% *w*/*w* and 7% *w*/*w* for sodium alginate and calcium chloride, respectively). The mean coating thickness predicted by the model was about 80 µm for a dipping time of 10 s, 85 µm for 120 s, 98 µm for 180 s, and 110 µm for 300 s, which is in good agreement with the experimental points collected at the same concentration of the coating solutions (Figure 5).

The corresponding linear regression equation describing the film thickness response (*R*^2^ = 0.95) as a function of dipping time alone is reported in Equation (11), for coded variables.
(11)$$Y_{1}\;=\;6.24\times 10^{-5}\;\times X_{1}-0.086$$

Regression analysis and ANOVA were used to assess the statistical significance of the different terms involved in the fitting model, which are shown in Table 4.

The statistical analysis confirms that the value of the coefficient *β*_1_ corresponding to the dipping time (*X*_1_) is smaller in value than the coefficients corresponding to the concentration of the coating solutions. It is also characterized by a *p*-value ≥ 0.1, confirming the statistically non-significant effect on film thickness. In contrast, both independent variables *X*_2_, and *X*_3_ (concentration of the sodium alginate and CaCl_2_ solutions, respectively) exhibited a significant effect on the thickness of the coating layer, with both *β*_2_ and *β*_3_ characterized by a *p*-value < 0.05. In addition, the interaction coefficient for the coating and cross-linking solution concentrations (*β*_23_) also resulted in having a significant effect on the response variable (*p* < 0.05), differently from the other interaction coefficients (*β*_12_ and *β*_13_), involving the dipping time variable (*p* ≥ 0.1). A more extensive statistical analysis is reported in the Appendix A.

Considering that the RSM analysis showed that it is not a critical parameter, the dipping time was set at 120 for subsequent experiments, because it ensured a more easily repeatable coating procedure than shorter times. The concentrations for the coating and cross-linking solutions were instead set at 0.5% *w*/*w* and 2.0% *w*/*w* for sodium alginate and calcium chloride, respectively, in subsequent experiments to ensure a thin and homogeneous coating layer.

The same formulation for the optimized alginate coating was used as a basis for the preparation of the active coating, incorporating the OEO nanoemulsion. The coating thickness, measured after cross-linking in the CaCl_2_ solution and drying, was only marginally affected by the addition of the nanoemulsion in the alginate solution. In the absence of nanoemulsion, the coating thickness on tomato skin was 4.8 ± 1.4 µm, whereas, in the presence of nanoemulsion, the coating thickness exhibited a statistically non-significant (*p* ≥ 0.05) reduction to 4.5 ± 2.7 µm.

### 3.5. Wettability of Tomato Peel with the Film-Forming Solutions

The effect of the addition of OEO nanoemulsions in the optimized coating solutions was investigated through contact-angle wettability analysis.

The contact angles of droplets of film solutions deposited on the tomato skin were lower than that of water (Figure 6). This suggests a high interaction of the film solutions with the hydrophobic tomato skin surface. The addition of OEO nanoemulsion to the alginate film-forming solution significantly (*p* < 0.05) decreased the contact angle value of about 14%, therefore further increasing the surface wettability by the active coating.

The contact angle values may range from 0° (complete and instantaneous spreading of the liquid onto the solid surface) to 180° (an unrealistic limit of absolutely no wetting) [40]. Therefore, the lower the contact angle, the higher the hydrophobicity of solutions. The observed reduction in contact angle in the presence of OEO might be attributed to nanoemulsion providing more hydrophobic sites in the coating, and therefore reducing water attraction. This behavior was observed from the moment the droplet was deposited on the tomato skin, with the contact angle quickly reaching an asymptotic value (details in Appendix A).

The surface wettability is an important factor in determining adhesion strength. Lower equilibrium contact angles correspond to increased adhesion strength. Figure 7 reports the adhesion strength, cohesive energy, and spreading coefficients of the coating solutions on the tomato surface in comparison with water.

The wettability of solid surfaces by coating solutions can be attributed to a mechanism involving the balance between adhesion and cohesion forces of the solution [34]. Adhesion and cohesion coefficients are, therefore, important parameters to determine the ability to spread over a surface. The adhesion force indicates the spreading ability of the liquid, and the cohesion force its contraction tendency [41]. The greater the values of the adhesion coefficient and the lower the cohesive coefficient, the better is the adhesion of the coating solutions on tomato skin.

No significant differences were observed between the adhesion coefficients of water and sodium alginate solution. On the contrary, the addition of OEO nanoemulsion caused a decrease in the adhesion coefficient. At the same time, the cohesion coefficients of both coating solutions were lower than that of water, with a reduction of −11.6% and −61.2% for sodium alginate and sodium alginate with OEO nanoemulsion, respectively. Furthermore, the surface tension decreased from 370.9 mN/m of water to 331.2 mN/m for sodium alginate solution and 145.5 mN/m for sodium alginate with OEO nanoemulsions. Altogether, these results show the highest wettability of hydrophobic nonpolar tomato surface for the sodium alginate solution containing the OEO nanoemulsions.

The spreading coefficient is one of the most important properties when evaluating the capacity of a solution to coat a surface of interest. The spreading coefficient (*W_s_*), calculated from Equation (8), is always negative, and values closer to zero indicate better surface wetting by the tested liquid [42]. Notably, the sodium alginate solution containing the OEO nanoemulsion exhibited higher values of the spreading coefficient. Likely, the addition of the nanoemulsion contributed to reducing the surface tension and to improve the adhesion on the tomato surface.

It must be remarked that the deposition of the alginate coating on the tomato surface, with or without OEO nanoemulsion, did not significantly affect the color of tomatoes, as shown by the pictures of coated samples in Appendix A and by the instrumental measurements reported in Appendix A of the Appendix A.

### 3.6. Effect of Coatings on Microbial Growth and Shelf Life of Tomatoes

Initial bacterial counts determined for uncoated and coated tomatoes ranged between 2.0 and 2.6 log CFU/g (Figure 8). No statistically significant differences were observed between the microorganism counts after one day of the storage period.

Total bacterial counts in uncoated samples increased slowly at the beginning, and then exponentially after three days of storage at room temperature. The microbial count became maximal between days 7 and 10, when it exceeded the acceptability limit for fresh tomatoes (6 log CFU/g; European Community Regulation n. 1441/2007). The samples coated with calcium chloride cross-linked alginate approached the acceptability limits during the same days; however, the presence of the coating contributed to a significantly (*p* < 0.05) slower microbial growth when compared to control. The use of the active coating, prepared with the OEO nanoemulsion, contributed to maintaining the total microbial count well below the acceptability limits, with values < 4.5 log CFU/g during the entire shelf life, and significantly lower (*p* < 0.05) than control and coating without OEO at days 7 and 10. The bacteriostatic action of the cross-linked alginate edible coating can be ascribed to the previously observed selective barrier properties against O_2_ and CO_2_ of alginate coatings [43]. Moreover, the obtained results show that the incorporation of OEO nanoemulsion greatly contributes to slowing down microbial growth.

Similarly to the aerobic mesophilic bacteria, coating significantly reduced the growth of yeasts and molds during storage compared to controls (Figure 9). Moreover, a significant difference between coated and control tomatoes was observed already from the first days of storage. During the entire storage period, molds and yeasts on coated tomatoes never exceeded 3 CFU/g, which was proposed as the limit for acceptability for nonthermal-processed fruit [44]. Notably, at the end of the shelf life, a reduced (of about 2.0 log cycles) yeast and mold population was observed for coated tomatoes in comparison with the control. These results are in line with those reported by other authors. Luksiene et al. 2012 reported inactivation levels between 1.0 and 1.3 log CFU/g of naturally distributed mesophilic bacteria in different fruit and vegetables, such as plum, cauliflower, sweet pepper, and strawberry [45]. Similar results have also been observed on other vegetable matrices, such as spinach, carrot, cabbage, and mushroom [46].

For molds and yeast, the incorporation of the OEO nanoemulsions in the coating caused statistically significant differences (*p* < 0.05) only from day 10.

The results of Figure 8 and Figure 9 show that the application of active edible coatings containing the OEO nanoemulsion contributed to further reducing the microbial growth on the tomato surface.

It must also be added that no significant impact on tomato color was observed during the entire shelf life, without any significant difference observed over time and between different groups (control tomatoes and tomatoes coated with alginate without and with OEO nanoemulsion), as shown in Appendix A of the Appendix A.

## 4. Conclusions

The deposition of an alginate coating layer, cross-linked with calcium chloride, was optimized as a function of the concentration of the coating solutions, finding that concentrations of 0.5% *w/w* for sodium alginate and of 2% *w/w* for calcium chloride were optimal for obtaining a thin (about 5 µm) and uniform coating layer on the tomato surface. The incorporation of an oregano essential oil (OEO) nanoemulsion in the optimized alginate coating solution did not significantly affect the resulting cross-linked coating thickness but improved tomato wettability because it conferred a more pronounced hydrophobic behavior to the solution. The active coatings prepared with OEO successfully reduced the growth of bacteria, yeasts, and molds on the surface of tomatoes stored for 14 days at ambient temperature, demonstrating that this could be a promising strategy to extend the shelf life of fresh fruit and vegetable during ambient-temperature storage.

## Figures and Tables

**Figure 1 foods-09-01605-f001:**
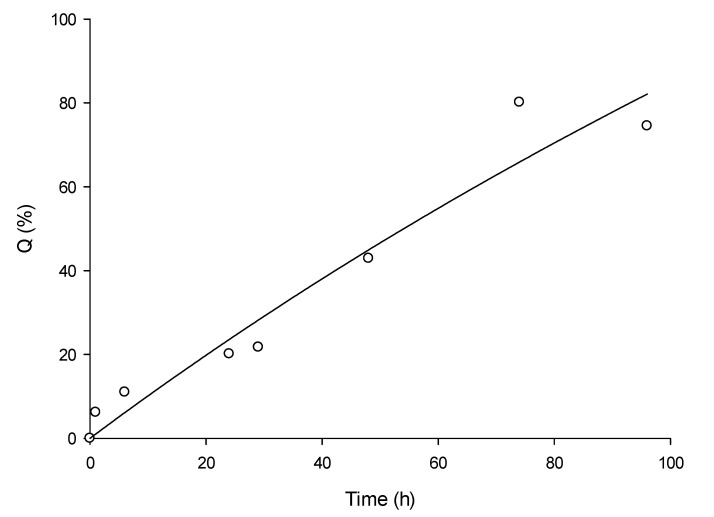
Cumulative release of OEO through cellulose membrane over time from the OEO nanoemulsion, normalized over the initial loading in the donor compartment.

**Figure 2 foods-09-01605-f002:**
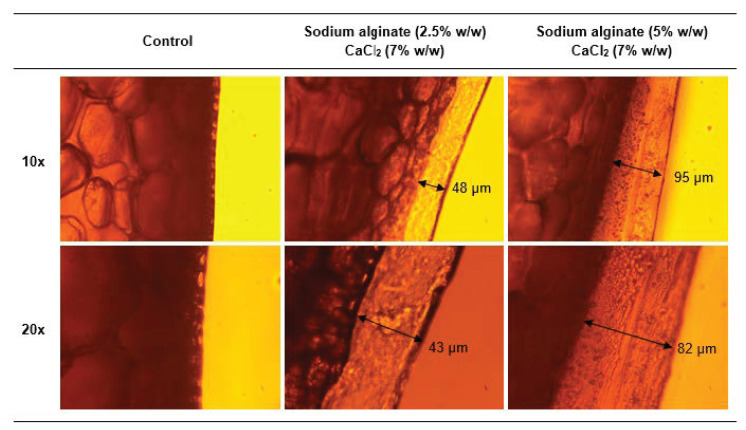
Microscopy images of uncoated and coated tomato peels prepared at different sodium alginate concentration and fixed dipping times (120 s in each solution), as described in the figure, and for two different magnifications (10× and 20×).

**Figure 3 foods-09-01605-f003:**
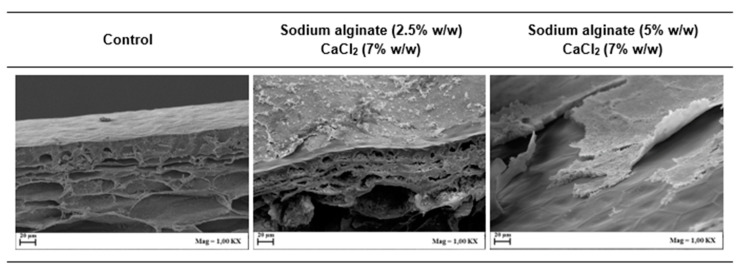
Scanning electron microscopy (SEM) of tomato peels without coating (control) and with sodium chloride coating at different sodium alginate concentrations at fixed dipping time (120 s).

**Figure 4 foods-09-01605-f004:**
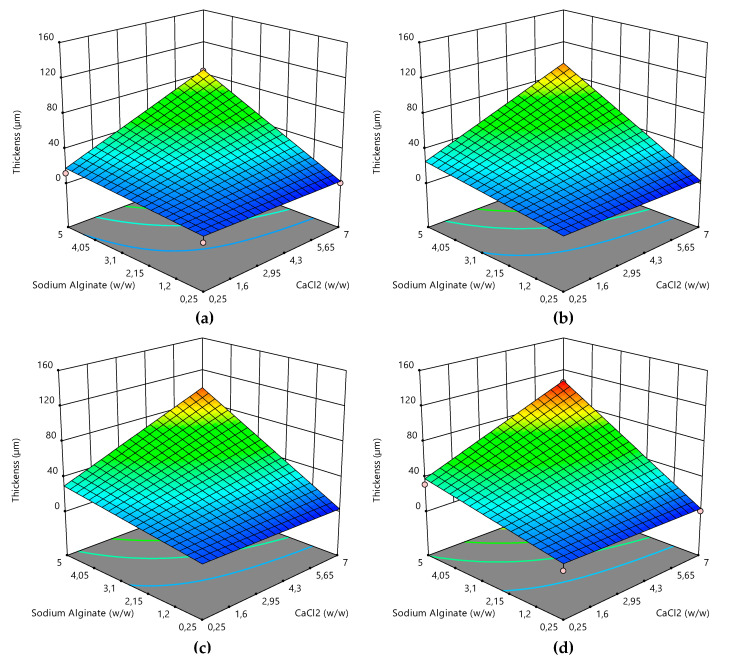
Response surface plots for the coating layer thickness obtained using sodium alginate coating solutions and calcium chloride cross-linking solutions of different concentrations for a dipping time of (**a**) 10 s, (**b**) 120 s, (**c**) 180 s, and (**d**) 300 s.

**Figure 5 foods-09-01605-f005:**
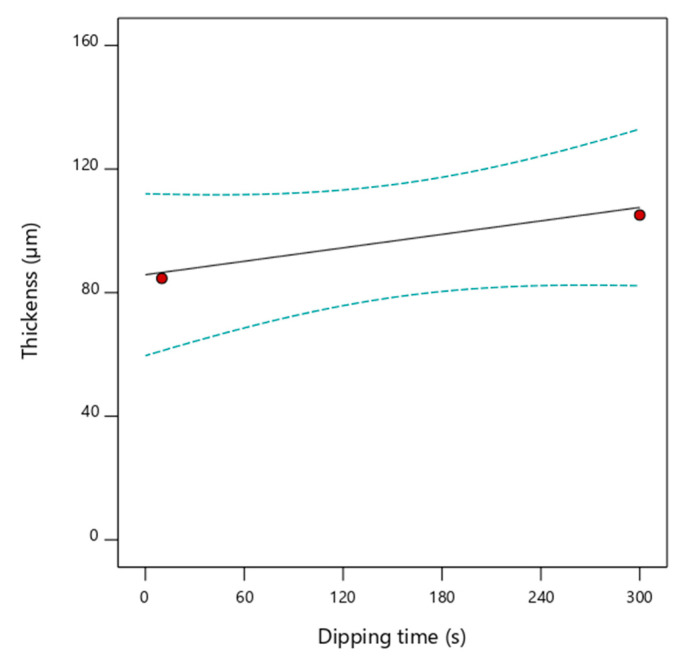
Dependence of coating thickness from dipping times (solid line) with the indication of the 95% confidence interval bands (dashed lines) and experimental design points (red circles), using sodium alginate at 5% *w*/*w* and CaCl_2_ at 7% *w*/*w* as coating solutions.

**Figure 6 foods-09-01605-f006:**
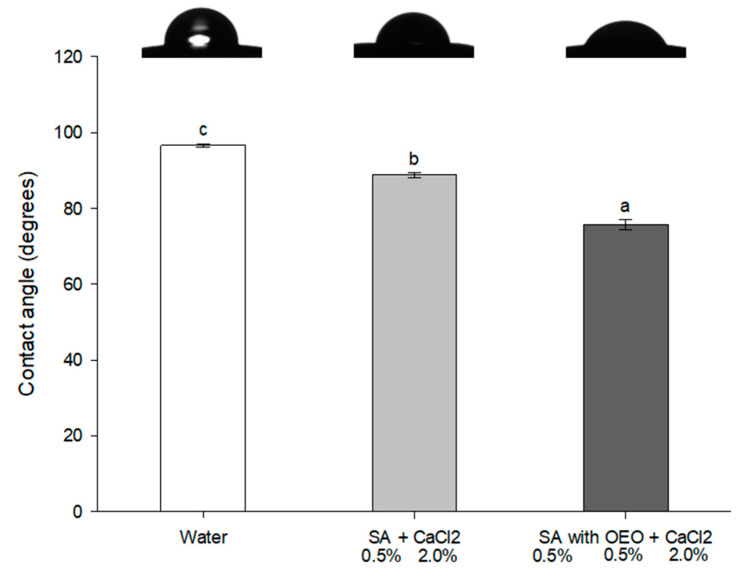
Average contact angle of water and coating solutions with or without OEO. Different letters denote significant differences (*p* < 0.05).

**Figure 7 foods-09-01605-f007:**
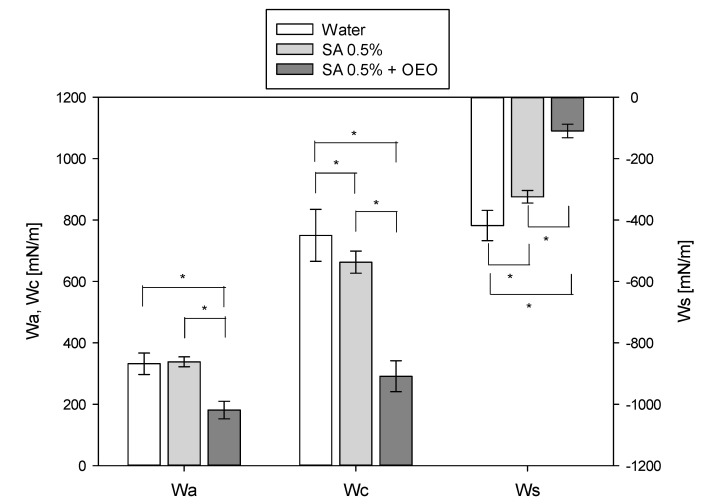
Adhesion (W_a_), cohesion (W_c_), and spreading (W_s_) coefficients of the alginate coating solutions without (identified by SA 0.5% in the legend) and with OEO nanoemulsion (identified by SA 0.5% + OEO in the legend) on tomato surface in comparison with pure water. Asterisks denote statistically significant differences (*p* < 0.05).

**Figure 8 foods-09-01605-f008:**
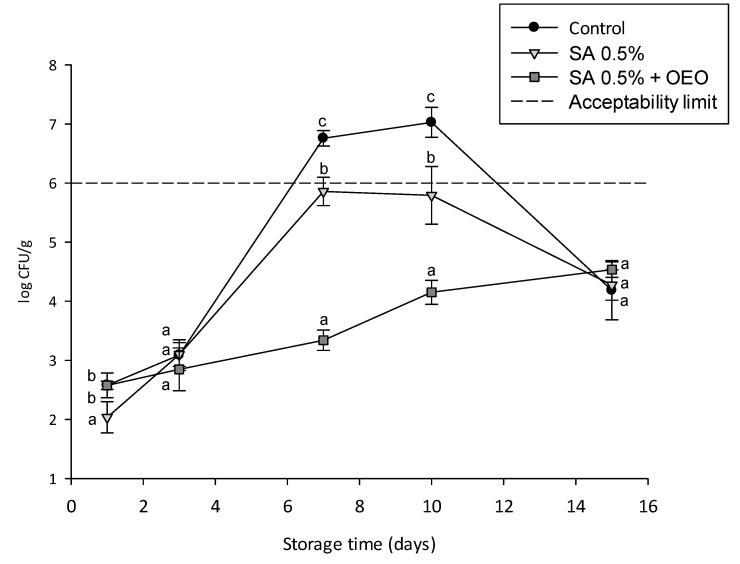
Total microbial count over a 14 days storage period at 24 ± 1 °C of uncoated tomatoes (control, black circles) and tomatoes with an edible coating without (SA 0.5%, light grey triangles) and with OEO nanoemulsion (SA 0.5% + OEO, grey squares). Different letters denote significant differences (*p* < 0.05) among the different samples for each day of storage.

**Figure 9 foods-09-01605-f009:**
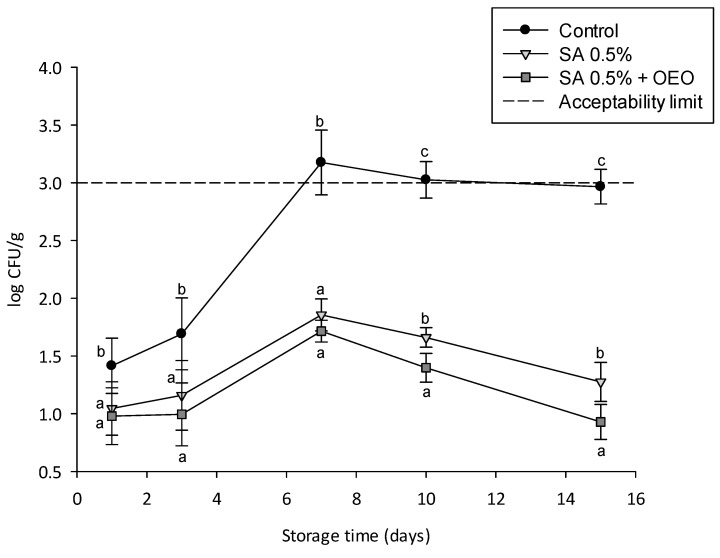
Yeast and mold count over a 14 days storage period at 24 ± 1 °C of uncoated (control, black circles) and tomatoes with an edible coating without (SA 0.5%, light grey triangles) and with OEO nanoemulsion (SA 0.5% + OEO, grey squares). Different letters denote significant differences (*p* < 0.05) among the different samples for each day of storage.

**Table 1 foods-09-01605-t001:** The central composite design used, with the indication of the actual value and the coded level (in parenthesis) of the three variables.

Standard Order	Run Number	Dipping Time (s) *X*_1_	Concentration (% *w*/*w*)	
			Calcium Chloride *X*_2_	Sodium Alginate *X*_3_
1	1	10 (−1)	0.250 (−1)	0.250 (−1)
13	2	155 (0)	3.625 (0)	0.250 (−1)
7	3	10 (−1)	7.000 (+1)	5.000 (+1)
10	4	300 (+1)	3.625 (0)	2.625 (0)
11	5	155 (0)	0.250 (−1)	2.625 (0)
18	6	155 (0)	3.625 (0)	2.625 (0)
14	7	155 (0)	3.625 (0)	5.000 (+1)
15	8	155 (0)	3.625 (0)	2.625 (0)
12	9	155 (0)	7.000 (+1)	2.625 (0)
5	10	10 (−1)	0.250 (−1)	5.000 (+1)
2	11	300 (+1)	0.250 (−1)	0.250 (−1)
4	12	300 (+1)	7.000 (+1)	0.250 (−1)
8	13	300 (+1)	7.000 (+1)	5.000 (+1)
19	14	155 (0)	3.625 (0)	2.625 (0)
3	15	10 (−1)	7.000 (+1)	0.250 (−1)
17	16	155 (0)	3.625 (0)	2.625 (0)
16	17	155 (0)	3.625 (0)	2.625 (0)
20	18	155 (0)	3.625 (0)	2.625 (0)
9	19	10 (−1)	3.625 (0)	2.625 (0)
6	20	300 (+1)	0.250 (−1)	5.000 (+1)

**Table 2 foods-09-01605-t002:** Formulation and physical characterization of oregano essential oil (OEO) nanoemulsion.

Composition		Hydrodynamic Diameter (nm)	PDI (-)	ζ-Potential (mV)
Oil Phase	Emulsifier			
*Origanum vulgare* essential oil 0.5% *w*/*w*	Lecithin 0.5% *w*/*w*	176.8 ± 2.4	0.26 ± 0.01	−23.7 ± 1.2

**Table 3 foods-09-01605-t003:** Fitting parameters and coefficients of determination for the fitting of the data of Figure 1, using Equation (10).

Parameters	Value
*a* _0_	1.0338
*b*	0.0022
*R* ^2^	0.9439
Adj *R*^2^	0.9346

**Table 4 foods-09-01605-t004:** Regression coefficients and probability *p*-values for thickness response.

Source	Coefficient	Sum of Squares	*p*-Value
β_0_	0.0339	0.0003	0.0001
β_1_	0.0054	0.0026	0.2373
β_2_	0.0161	0.0083	0.0027
β_3_	0.0287	4.88 × 10^−8^	<0.0001
β_12_	0.0001	0.0002	0.9875
β_13_	0.0050	0.0027	0.3193
β_23_	0.0184	0.0025	0.0024
Ɛ		0.0025

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
