# Peer review of "Edible Coatings Containing Oregano Essential Oil Nanoemulsion for Improving Postharvest Quality and Shelf Life of Tomatoes"

_foods, 2020, doi:10.3390/foods9111605_

Round 1

Reviewer 1 Report

The manuscript deals with the edible coatings containing oregano essential oil nanoemulsion for improving postharvest quality and shelf life of tomatoes.

The English language must be revised.

Abstract

Please present your main results.

Materials and methods

Color analysis?texture determination?

Results and discussion

Pictures of each sample?

Figure 2- why showing two different magnification levels with different thicknesses?if they are the same samples.

Figure 3- Results seem to be from different areas?this analysis shows some artifacts and it is not totally clear.

References

Please format scientific names in italic.

Around 25 references have more than 5 years. Please update your list of references.

Author Response

Dear reviewer 1,

thank you for your valuable contribution to improve our manuscript. We have made all the suggested corrections, as described in details in the attached file.

Best regards,

Francesco Donsì

Reviewer 2 Report

In the study, reported in this manuscript, the authors intended to improve tomato storability by the application of calcium chloride cross-linked alginate-based edible coatings, with or without the addition of oregano essential oil (OEO) nanoemulsion as a natural antimicrobial. The authors initially optimised the coating formulations, provided a detailed analysis of the physical properties of the resulting coatings and evaluated their effects on tomato shelf life by analyses of changes in total microbial loads, and yeasts and molds counts over 14 d of storage (at room temperature and not refrigerated, line 23). Research on edible coatings is currently very attractive and the topic of the manuscript fits the scope of Foods.

The text is generally well-written, though many sentences are rather wordy and should be polished. In addition, the manuscript contains quite a few minor problems that need to be solved in a moderate revision.

General comments

Keywords: As electronic search engines focus on both Title and Keywords, it is better not to repeat words already used in the title. This increases the chance to get listed.

Many of the sentences are wordy, long and complex, and not always easy to understand. Try to keep them short and simple (kiss), which would facilitate reading.

Please always replace the term “weight” by “mass”. Although still very often used instead of mass, not only in everyday language, physically correctly weight denotes a force, given in N, which is derived from mass (Fw = m * g). See relevant books of physics or engineering, or, simply, http://en.wikipedia.org/wiki/Mass.

The plural of fruit is fruit, except for reference to different species, i.e. apples, cherries and peaches altogether are fruits.

SI requires that numerals be followed by proper SI units; e.g. 12 d, not 12 days (min for minute(s), h for hour(s), s for second(s)). Likewise words should not be followed by unit abbreviations; e.g., twelve days, not twelve d.

Specific comments

30-36 These are meanwhile commonplaces that have been presented too often. Your introduction does not need this. Better start directly with the next paragraph.

38 “traditionally used in food conservation”??? Why food conservation in this context?

49-50 “including spraying, dipping, spreading and thin film hydration [8].”

56 Better start a new paragraph.

65 Why not simply “which can also form thin films or coatings”. Active voice.

68-70 Why not simply “For example, dipping fruits and vegetables in calcium chloride solutions decreased postharvest softening and decay if combined with heat treatment [20].“

73-74 Why not simply “Oregano essential oil (OEO), which is rich in the GRAS monoterpene carvacrol, has excellent antimicrobial properties and inhibits the growth of both gram-positive and gram-negative bacteria and of fungi [24].”

87 Solanum lycopersicum and cerasiforme should be italicized but not “var.”

96 “Germany) and lecithin”

105 “at 18,000 rpm for 5 min”

117 “Malvern Instruments Ltd., Malvern, United”

124 “sources at 25 °C.”

136 “results are means of two replicated measurements.”

163 “sterilized in autoclave at 121 °C for 10 min.”

165 “at a mass ratio of 2:1,”

173 “immersion of the fruit”

208 “thickness. Eight measurements” Please do not start a sentence with a number.

216/222 “Slices, 2-mm thick, “

256-257 “both control and coated tomatoes stored at 24 ± 1 °C for 14 d were determined for total bacterial count and those of yeasts and molds at storage days 1, 3, 7, 9 and 14.”

263 “two tomatoes per coating formulation were analyzed”

274 “Treatments and analyses”

275 “as means ± sd.”

276 “Significance of differences between means were evaluated”

277 SPSS is owned by IBM in 2009.

284 “systems [38].”

287-290 “Therefore, the obtained nanoemulsion did not show any sign of instability (e.g. creaming or oiling off) and always maintained a homogeneous appearance over a 1-month storage period at 4 °C. The nanoemulsion is likely stabilized by a combined effect of electrostatic and steric effects.” Kiss

296 “compartment continuously increased from 6.2% after 1 h to 74.5% after 96 h (Figure 1).” Kiss

302-304 This is largely MaM!

355 “solutions (Figure 5).” Kiss

366-380 Please concentrate on the analysis of the true and relevant effects and do not stress the statistics so much. The journal is Foods not Statistics.

395 “The contact angles of droplets of film solutions deposited on the tomato skin were lower than that of water (Figure 6). This suggests a high interaction of the film solutions with the hydrophobic tomato skin surface.” Kiss

439-442 No, Figure 8 & 9 only show the total microbial (8) and yeast and mold (9) counts. Only the analysis of the presented results provides information on the potential antibacterial activities of the respective coatings. Furthermore, the sentences do not really add at all and may be deleted. Kiss

444 “2.6 log CFU/g (Figure 8).”

445 “which confirmed that the initial microbial load was the same for all samples.” I hope so, otherwise you experimental approach would have been insufficient.

446-447 Please delete. This sentence just extends the manuscript, but did not add anything.

449-452 Please also always explain the symbols in the caption of the figures.

454-456 “The microbial count became maximal between days, when it exceeded the acceptability limit for fresh tomatoes (6 log CFU/g; EC Regulation n. 1441/2007).” kiss

456-457 “For coated samples, the acceptability limit was never exceeded.” This is only partially correct; see “+sd” of SA 0.5 % on day 7 and 10. In the figure 8, it is certainly not day 9.

463-466 Please rewrite this sentence. Kiss

467-471 “Similarly to the aerobic mesophilic bacteria, coating significantly reduced the growth of yeasts and molds during storage, compared to controls (Figure 9).” This is all you need. Kiss

471-473 Please rewrite this sentence. Kiss

489-492 This is repeated in 500-503. One time should be enough.

Author Response

Dear reviewer 2,

thank you for your valuable contribution to improve our manuscript. We have made all the suggested corrections, as described in detail in the attached file.

Best regards,

Francesco Donsì

Round 2

Reviewer 1 Report

The manuscript was improved.